# *Wikstroemia**ganpi* Extract Improved Atopic Dermatitis-Like Skin Lesions via Suppression of Interleukin-4 in 2,4-Dinitrochlorobenzene-Induced SKH-1 Hairless Mice

**DOI:** 10.3390/molecules26072016

**Published:** 2021-04-01

**Authors:** Jonghwan Jegal, No-June Park, Beom-Geun Jo, Tae-Young Kim, Sim-Kyu Bong, Sangho Choi, Jin-Hyub Paik, Jung-Won Kim, Su-Nam Kim, Min Hye Yang

**Affiliations:** 1College of Pharmacy, Pusan National University, Busan 46241, Korea; jhjegal@pusan.ac.kr (J.J.); bg_jo@pusan.ac.kr (B.-G.J.); taeyour@pusan.ac.kr (T.-Y.K.); 2Natural Products Research Institute, Korea Institute of Science and Technology, Gangneung 25451, Korea; 115519@kist.re.kr (N.-J.P.); 115044@kist.re.kr (S.-K.B.); 3International Biological Material Research Center, Korea Research Institute of Bioscience and Biotechnology, Daejeon 34141, Korea; decoy0@kribb.re.kr (S.C.); jpaik@kribb.re.kr (J.-H.P.); 4Department of Costmetology, Changshin University, Paryong-ro, Masanhoewon-gu, Changwon-si 51352, Korea; won104@cs.ac.kr

**Keywords:** *Wikstroemia ganpi*, atopic dermatitis, 2,4-dinitrochlorobenzene, immunoglobulin E, interleukin-4

## Abstract

Plants of the genus *Wikstroemia* are used in Chinese traditional medicine to treat inflammatory diseases, such as arthritis, bronchitis, and pneumonia. The present study was designed to determine whether *Wikstroemia ganpi* (Siebold and Zucc.) Maxim. offers a potential means of treating 2,4-dinitrochlorobenzene (DNCB)-induced atopic dermatitis (AD) in mice. Symptoms such as redness, edema, and keratinization in AD mice induced by DNCB were alleviated by the co-application of an ethanolic extract of *W. ganpi* for 2 weeks. The severity of skin barrier function damage was evaluated by measuring TEWL (transepidermal water loss). TEWLs of DNCB sensitized mouse dorsal skin were reduced by the application of a *W. ganpi* ethanolic extract, and skin hydration was increased. In addition, the infiltration of inflammatory cells into the dermis was significantly reduced, as were blood levels of IgE and IL-4, which play an important role in the expression of AD. The results of this experiment suggest that *W. ganpi* is a potential therapeutic agent for AD.

## 1. Introduction

Atopic dermatitis (AD) is a multifactorial, common inflammatory skin disease characterized by representative symptoms, such as pruritus, dry skin, and lichenification. AD is classified as extrinsic or intrinsic AD depending on the presence or absence of immunoglobulin E (IgE)-mediated responses by external allergens. The majority of AD patients (~80%) have extrinsic AD with an IgE response. Immunological features in extrinsic AD patients include high IgE blood levels, Th2 cell proliferation, and high blood levels of cytokines (e.g., IL-4) produced by Th2 cells. Degranulation by basophils also increases blood levels of histamine [1,2]. Immune responses involving Th2 cells are known to weaken the skin barrier. The cytokines secreted by these cells (IL-4 and IL-13) downregulate components of the skin barrier, molecules involved in cell junctions, and ceramides [3]. Diminished skin barrier function can result in the invasion of harmful exogenous substances, and loss of skin hydration is the one of the main drivers of the development of extrinsic AD [4]. Furthermore, ineffective prevention of allergens’ invasion due to skin barrier damage results in an allergen-induced inflammatory response. Transepidermal water loss (TEWL) provides a means of measuring skin barrier function [5,6], as TEWL values increase in proportion to the severity of AD. In patients with intrinsic AD, TEWL values are at normal levels [7]. In these patients, two mechanisms play key roles in the development of AD; that is, Th2 cells and skin barrier damage, though, notably, these two factors affect each other and establish a vicious cycle. Accordingly, AD patients are treated with topical corticosteroids or calcineurin inhibitors to suppress immune response, and emollients to strengthen the skin barrier [8,9].

Flavonoids are one of the most abundant polyphenolic secondary metabolites synthesized by plants. Structurally, they possess of a three-ring phenyl benzopyrone structure (C6-C3-C6), and, in nature, exist as free aglycones or glycosides. Depending on the number and locations of oxygens on the middle pyrone ring and on substitutions, flavonoids are classified as anthocyanins, flavans, flavones, flavanols, flavonols, flavanones, flavonones, isoflavones, and others [10,11,12,13]. Flavonoids have broad spectrum biological activities, which include anti-inflammatory, antithrombotic, anticancer, vasodilatory, antihepatotoxic, and antiosteoporotic effects [14,15]. Recently, a number of authors have suggested flavonoids might be potentially useful treatments for AD, based on considerations of their anti-inflammatory and antiallergic properties [16,17,18,19,20].

*Wikstroemia ganpi* (Siebold and Zucc.) Maxim. is a deciduous shrub of genus the *Wikstroemia* of the Thymelaeaceae family, and is endemic in Australia, Japan, and the southern region of South Korea [21]. Plants of the genus, *Wikstroemia*, have traditionally been used as source for making paper in East Asia. Furthermore, *W. indica* is used in folk medicines to treat cancer [22], whooping cough, arthritis, and syphilis [23], under the name “liaogewang” in China [23,24,25]. Currently, a product containing a water extract of *W. indica* root is sold over the counter in China for the treatment of various inflammatory diseases, such as bronchitis, tonsillitis, and mumps [26]. Various bioactive substances, including flavonoids, coumarins, and lignans, have been reported in plants of the *Wikstroemia* genus [27], and many studies have described the anti-inflammatory, anticancer, antifungal, and antiviral activities of its extracts [22,26,28]. However, comparatively little is known of the bioactivities of *W. ganpi*, and, in particular, no attempt has been previously made to evaluate its effect on skin barrier function or anti-atopic activity. The present study was performed to evaluate the anti-atopic activity of an ethanolic extract of *W. ganpi* (WGE) in DNCB (2,4-dinitrochlorobenzene)-induced murine models of AD, and to document the phytochemicals present in the plant.

## 2. Materials and Methods

### 2.1. Plant Materials and Extraction

Aerial parts of *Wikstroemia ganpi* (Siebold and Zucc.) Maxim. were collected in Geumsa-ri, Yeongnam-myeon, Goheung-gun, and Jeollanam-do, Republic of Korea, and identified by Jin-Hyub Paik (International Biological Material Research Center, Korea Research Institute of Bioscience and Biotechnology). A voucher specimen (PNU-0027) was deposited in the College of Pharmacy, Pusan National University. *W. ganpi* samples (21 g dry plant) were extracted twice with 2 L volumes of 95% EtOH for 48 h, filtered, and filtrates were combined and evaporated in vacuo at 35 °C to give a semisolid residue, which was then freeze-dried to produce a *W. ganpi* EtOH extract (WGE) of 2.6 g and 9.3% yield.

### 2.2. Animals

The animals used were 6-week-old female SKH-1 hairless mice, which were procured from Orient Bio Inc. (Seongnam, Korea). Animals were used for experiments after a 7-days acclimatization period in our animal laboratory, which was maintained at 25 ± 5 °C and 55 ± 5% RH under a 12-hour light and dark cycle. Animals were provided ad libitum with standard animal chow and water. The guidelines for all animal experiments were in compliance with the Guide for the Care and Use of Laboratory Animals of the National Institutes of Health (2013), and were approved by the Institutional Animal Care and Use Committee (IACUC) of Korea Institute of Science and Technology (Certification No. KIST-2016-011, 2016).

### 2.3. Atopic Dermatitis-Induced Mice Model by DNCB Treatment

To induced atopic dermatitis, DNCB (2,4-dinitrochlorobenzene, Sigma-Aldrich, Seoul, Korea) was used and diluted at 1% and 0.1% in a 7:3 mixture of propylene glycol and ethanol. SKH-1 hairless mice that had been acclimatized for 1 week were topically applied 1% DNCB (100 μL) once a day for 7 days (days 1 to 7). After DNCB sensitization, mice were resensitized with 0.1% DNCB (100 μL) three times a week for 2 weeks. Negative controls were treated by DNCB resensitization and vehicle (CON group), animals in the treatment group received DNCB resensitization and 1% *W. ganpi* EtOH extract (*W. ganpi* group), and animals in the positive control group received DNCB resensitization and 1% pimecrolimus (Elidel group) twice a day for 2 weeks. The dermatitis score was evaluated by scoring the skin lesions according to symptoms, such as erythema/hemorrhage, scarring, dryness, edema, and excoriation/erosion. The severity of dermatitis in skin was assessed using the Eczema Area and Severity Index scoring system: 0, no symptoms; 1, mild symptoms; 2, moderate symptoms; and 3, severe symptoms. On the day 22, mice were sacrificed and dorsal skin tissues were excised for histological examination, blood was collected from the abdominal aorta, and serum IgE and IL-4 levels were measured.

### 2.4. Measurement of Transepidermal Water Loss (TEWL) and Skin Hydration

A Tewameter TM210 (Courage and Khazaka, Cologne, Germany) and a SKIN-O-MAT (Cosmome, Rhur, Germany) unit were used to measure transepidermal water loss (TEWL) and skin hydration, respectively, of dorsal skin, according to the manufacturer’s instructions. Measurements were recorded weekly under ambient conditions (25 ± 5 °C, RH 55 ± 5%).

### 2.5. Measurement of Total Serum IgE and IL-4 Levels

Blood samples were collected from the abdominal aorta of SKH-1 hairless mice and serum were collected by centrifugation at 10,000 rpm for 15 min at 4 °C. Total serum IgE and IL-4 levels were measured using enzyme-linked immunosorbent assay kits (eBioscience, San Diego, CA, USA), according to the manufacturer’s instructions. 

### 2.6. Real-Time Quantitative Polymerase Chain Reaction (q-PCR) 

To isolate total RNA, the mice dorsal skin tissue was homogenized, and isolated using the RNeasy Mini Kit (Qiagen, Valencia, CA, USA), according to the manufacturer’s instructions. The isolated RNA was reverse-transcribed using the cDNA synthesis kit (Thermo Fisher Scientific Bremen, Germany). PCR was performed using the Power SYBR^®^ Green PCR Master Mix (Applied Biosystems, Foster City, CA, USA) under the following thermal cycling conditions: 95 °C for 2 min; 95 °C for 5 sec, and 59 °C for 30 sec, for a total of 40 cycles. The primer sets were as follows: INF-γ, forward 5′- GTC ACA GTT TTC AGC TGT ATA GGG -3′ and reverse 5′- AGC GGC TGA CTG AAC TCA GAT TGT A -3′; TNF-α, forward 5′- AGC CCC CAG TCT GTA TCC TT -3′ and reverse 5′- CTC CCT TTG CAG AAC TCA GG -3′; IL-4, forward 5′- ACC TTG CTG TCA CCC TGT TC -3′ and reverse 5′- TTG TGA GCG TGG ACT CAT TC -3′; GAPDH, forward 5′-ACC ACA GTC CAT GCC ATC AC-3′ and reverse 5′-CCA CCA CCC TGT TGC TGT A-3′. Data analyses were performed on QuantStudio™ 6 Pro System (Applied Biosystems). Quantification of gene expression with Q-PCR data was determined relative to GAPDH.

### 2.7. Histological Examination

For histological analysis, dorsal skin tissue was first detached, fixed in a 10% formalin solution, and embedded in paraffin wax for 24 h. Embedded tissues were sectioned at 2–3 mm, and Hematoxylin and eosin (H&E) and toluidine blue staining were performed to measure changes in epidermal thicknesses and mast cell counts. Stained tissues were observed and photographed under an optical microscope (Olympus CX31/BX51, Olympus Optical Co., Tokyo, Japan) and a fluorescence microscope (TE2000-U, Nikon Instruments Inc., Melville, NY, USA).

### 2.8. High Performance Liquid Chromatography Photodiode Array (PDA) Detector Analysis

Chemical profiling of main constituents in the *W. ganpi* extract were determined with HPLC-PDA. The HPLC-PDA analysis was performed using a Waters e2695 separation module equipped with a 2998 photodiode array (PDA) detector (Waters Corporation, Milford, MA, USA) and a SunFire^®^ C18 column (4.6 × 250 mm^2^, 5 μm, Waters) maintained at 30 ± 2 °C. Solvent A (0.1% formic acid-acetonitrile) and solvent B (0.1% formic acid-water) were used as the mobile phases (0–5 min, 5–10% A; 5–20 min, 10–30% A; 20–25 min, 30–50% A; 25–30 min, 50% A). The injection volume was 10 µL, the flow rate was set at 1.0 mL/min, and detection was performed at 340 nm.

### 2.9. Statistical Analysis

Statistical analysis was performed by one-way analysis of variance (ANOVA) and Tukey’s multiple comparisons post-hoc analysis. All experiments were performed independently, at least twice. Results are expressed as means ± standard errors of means (SEMs), and *p* values of < 0.05 were considered statistically significant.

## 3. Results

### 3.1. Effects of Wikstroemia ganpi Extract on AD-Like Lesions in the DNCB-Induced Mouse Model

To assess the anti-atopic effect of WGE in the DNCB model, typical AD-like skin symptoms, such as erythema, excoriation, xeroderma, and exudation were evaluated when DNCB was administered for three weeks. A schematic of the experiment procedure is provided in Figure 1a. In the WGE treatment group (*W. ganpi* group), AD symptoms were noticeably alleviated as compared with the negative control group (Figure 1b,c). H&E staining was performed to determine epidermal thicknesses and toluidine staining was used to observe mast cell infiltration. Epidermal thickness and mast cell infiltration in the DNCB group were about 2.5 times and 2.8 times greater, respectively, than in the CON group. In the *W. ganpi* group, epidermal thickness was about 37.3% thinner (47.20 μm) than in the DNCB group (75.25 μm) (Figure 2a,b), and the number of mast cells that migrated to the dermis was significantly reduced by about 42.3% (Figure 3a,b). In the positive control-treated group (Elidel group), epidermal thickness was about 16.2% thinner (59.04 μm) than in the DNCB group (75.25 μm). The number of mast cells infiltrated to the dermis in the Elidel group was significantly reduced by about 23.9% in comparison to the DNCB group. As a result, *W. ganpi* treatment showed a better effect on both epidermal thickness and mast cells infiltration compared to Elidel treatment.

### 3.2. Effects of Wikstroemia ganpi Extract on Blood Serum IL-4 and IgE Levels

Serum was collected from DNCB-treated SKH-1 hairless mice to measure IL-4 and IgE levels, which are characteristically elevated in AD patients. In the DNCB group, serum concentrations of IL-4 and IgE were significantly higher by 4.98- (64.90 pg/mL) and 3.76- (262.83 ng/mL) fold, respectively, as compared with the 17.25 pg/mL and 52.75 ng/mL observed in the CON group. DNCB-induced serum IL-4 levels were inhibited by 1% WGE treatment to about 32.4% (43.85 pg/mL) (Figure 4a). Treatment with 1% WGE also reduced the serum IgE level to about 44% (146.93 ng/mL) (Figure 4b). Treatment with Elidel lowered serum IL-4 concentration to about 44.7% (35.88 pg/mL) compared to the DNCB group, which was similar to that of the 1% WGE treatment group. However, no significant decrease in serum IgE in the Elidel group was detected.

### 3.3. Effects of WGE on mRNA Expression of TNF-α, IFNγ, and IL-4 in the Back Skin

The effect of WGE on the mRNA expression of pro-inflammatory cytokines was investigated in the the dorsal skin of the SKH-1 hairless mice. The application of DNCB for 21 days in the DNCB group considerably increased mRNA expression of TNF-α, IFN*γ*, and IL-4, as compared with the CON group, whereas WGE treatment significantly reduced DNCB-induced expression of the proinflammatory cytokines (Figure 5). The expression of TNF-α, IFN*γ*, and IL-4 mRNA in the WGE group was significantly reduced by about 80%, 86%, and 71%, respectively, when compared to the DNCB group.

### 3.4. Effects of Wikstroemia ganpi Extract on Skin Barrier Function in the DNCB-Induced Mouse Model

DNCB-induced skin barrier damage increased TEWL and reduced skin hydration. TEWL greatly increased from 29.48 J (g/m^2^/h) to 91.83 J (g/m^2^/h) after the 7-days DNCB sensitization period, after which it decreased significantly to 20.0% after 1% WGE and DNCB were co-administered for 14 days (Figure 6a). DNCB application also decreased the value of skin hydration to 27.63% compared to the CON group after the 7-days sensitization (Figure 6b). On day 22, skin hydration levels of the 1% WGE-treated mice significantly restored up to 21.03%. Elidel treatment lowered TEWL to 17.5% compared to the CON group on day 22, and increased the skin hydration level to 21.23% compared to the CON group (Figure 6a,b).

### 3.5. HPLC-PDA Analysis Result of WGE

Chromatographic profile results indicate that WGE contained two coumarins and three flavonoids. The retention time (*t*_R_) and UV absorption maxima (λ_max_) of each peak are as follows: (1) 7-methoxyluteolin-5-*O*-glucoside (*t*_R_ 20.159 min, λ_max_ 241.3/341.4 nm); (2) quercitrin (*t*_R_ 21.245 min, λ_max_ 255.5/348.6 nm); (3) pilloin 5-*O*-β-d-glucopyranoside (*t*_R_ 23.119 min, λ_max_ 242.5/340.2 nm); (4) triumbelletin (*t*_R_ 27.888 min, λ_max_ 327.0 nm); and (5) daphnoretin (*t*_R_ 28.531 min, λ_max_ 343.8 nm) (Figure 7).

## 4. Discussion

AD is a chronic inflammatory skin disease caused by a variety of interacting factors. Although the pathogenesis of AD is unclear, recent research has shown that skin barrier damage contributes significantly to the development of AD. Furthermore, increased levels of Th2 cell-related inflammatory cytokines, such as IL-4 and IgE, are invariably observed in AD patients [3,29]. Accordingly, topical corticosteroids or calcineurin inhibitors that inhibit excessive immune activity and/or moisturizing agents containing ceramide or hyaluronic acid, which enhance skin barrier function, are used in the management of AD [30]. However, topical corticosteroids, which are considered first-line medical therapies, cannot be used for a long time due to side effects, such as skin atrophy, striae, and telangiectasis [30]. On the other hand, calcineurin inhibitors can be used long-term, but the range of their indications is relatively narrow [31]. For these reasons, there is increasing demand for alternative agents for the treatment of AD.

Many studies have shown that plants of the genus *Wikstroemia* have anti-inflammatory activity [27,32,33]. However, no systematic studies have been conducted to evaluate the anti-atopic activity of *W. ganpi*. Our prescreening experiments revealed that WGE inhibits the production of IL-4 in RBL-2H3 cells, and, thus, based on this result, we conducted the present study to determine whether WGE has a therapeutic effect in AD. In our DNCB model, allergic reactions similar to AD, such as skin thickening, redness, and dryness, and exudates were observed. In addition, TEWL and skin hydration testing showed DNCB reduced skin barrier function and degree of skin hydration. However, when 1% of WGE was co-applied with DNCB, these effects were significantly alleviated. Weakening of skin barrier function facilitates allergen penetration into the epidermis and induces excessive immune response [9,34]. For this reason, in mild-to-moderate AD, non-pharmacological treatments based on emollients containing ingredients that retain moisture, such as ceramide, are used to alleviate AD symptoms [8]. The present study shows that WGE also acts as a moisturizing agent and maintains skin barrier function, and, thus, alleviates the symptoms of AD.

Increased blood levels of inflammatory cytokines, IL-4 and IgE, due to activation of Th2 cells, are one of two axes of the most characteristic AD symptoms, and their upregulation by DNCB was also reduced by WGE in our murine model. When an allergen penetrates the epidermis, Langerhans cells promote the differentiation of Th2 cells and a Th2 cell-mediated immune response is initiated. Cytokines produced by Th2 cells, such as IL-4 and IL-13, are known to induce isotype switching for IgE production [35]. Based on our observations, it seems that the weakening of an allergic reaction due to inhibitions of the productions of IL-4 and IgE by WGE ameliorated the symptoms of AD. Recent studies have shown that IL-4 increases are associated with the inhibition of skin barrier function. Ceramide, which plays an important role in maintaining skin barrier function, is produced by enzymes that activate TNFα- and IFNγ-induced transcriptions, and the enzymes are inhibited by IL-4. That is, excessive production of IL-4 weakens skin barrier function and, when external allergen invasion increases due to a weakened skin barrier, a Th2 cellular immune response reoccurs and a vicious cycle is established, leading to excessive production of IL-4 [9,36].

## 5. Conclusions

In conclusion, this study shows that WGE ameliorated the symptoms of AD in our DNCB-induced mouse model. WGE improved skin barrier function, reduced TEWL and increased skin moisture contents. In addition, the application of WGE reduced serum levels of IL-4 and IgE in AD mice, thereby weakening the allergic immune response caused by AD. Our findings indicate WGE has potential use a therapeutic agent for AD.

## Figures and Tables

**Figure 1 molecules-26-02016-f001:**
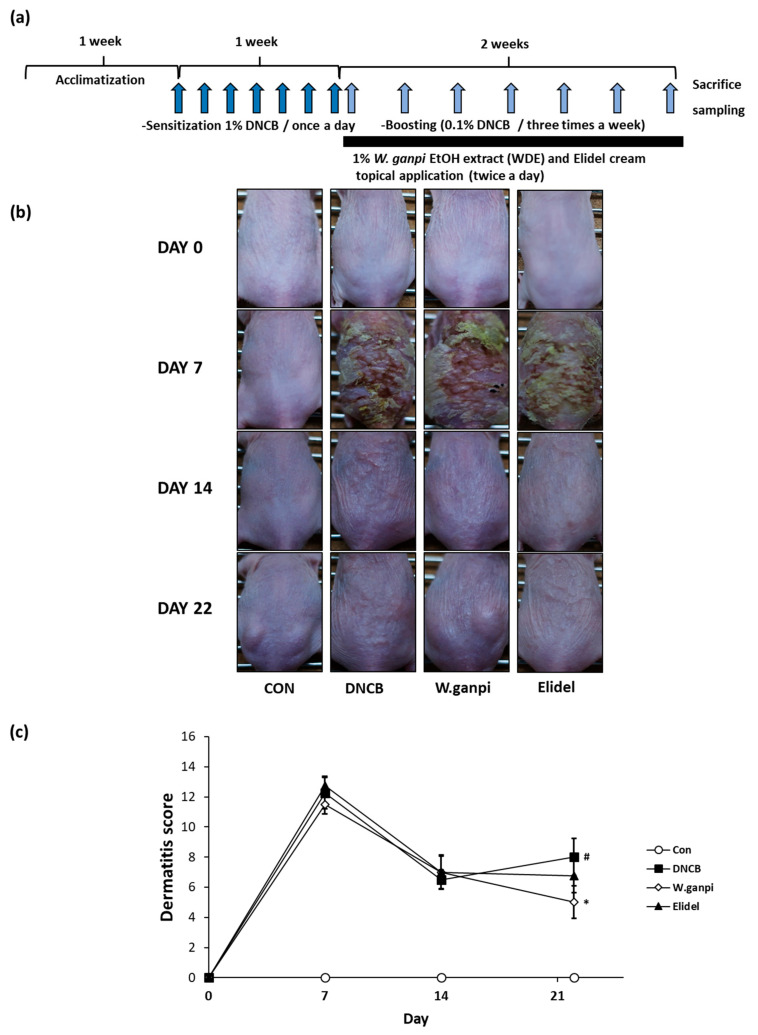
Effects of *W**. ganpi* EtOH extract (WGE) on the development of Atopic dermatitis (AD)-like symptoms induced by 2,4-dinitrochlorobenzene (DNCB) in the dorsal skin of the SKH-1 hairless mice. (**a**) Summarized diagram of the experiment schedule; (**b**) Clinical change of the AD-like skin lesion; (**c**) Dermatitis score. CON: control group, DNCB: DNCB-treated group, *W. ganpi*: DNCB and 1% *W. ganpi* EtOH extract-treated group, Elidel: DNCB and Elidel cream treated group. ^#^
*p* < 0.05 vs. the CON group; * *p* < 0.05 vs. the DNCB group.

**Figure 2 molecules-26-02016-f002:**
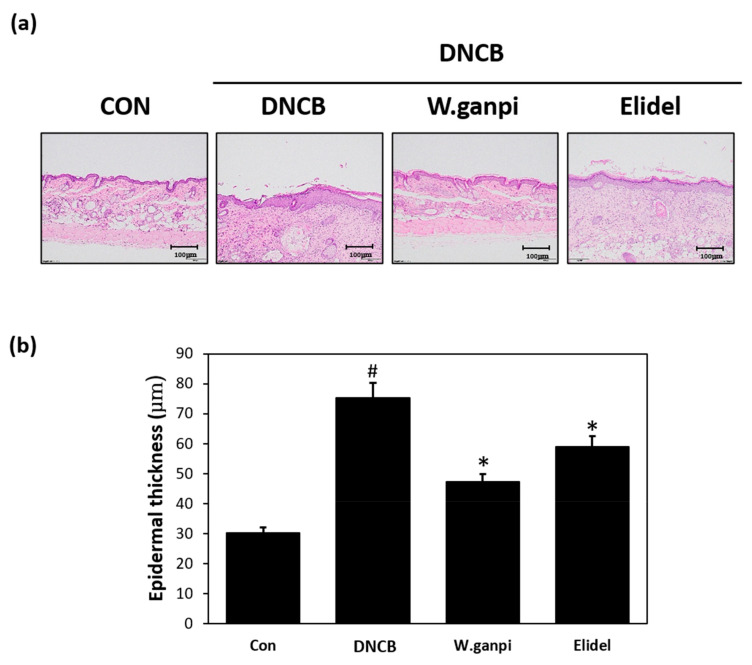
Effects of WGE on the change of the epidermal thickness in AD-like symptoms induced by DNCB in hairless mice. (**a**) Results of changes in epidermal thickness of skin lesions confirmed by histopathological examination. Stained by Hematoxylin and eosin (H&E) staining; (**b**) Epidermal thickness. CON: control group, DNCB: DNCB-treated group, *W. ganpi*: DNCB and 1% *W. ganpi* EtOH extract-treated group, Elidel: DNCB and Elidel cream treated group. ^#^
*p* < 0.05 vs. the CON group; * *p* < 0.05 vs. the DNCB group.

**Figure 3 molecules-26-02016-f003:**
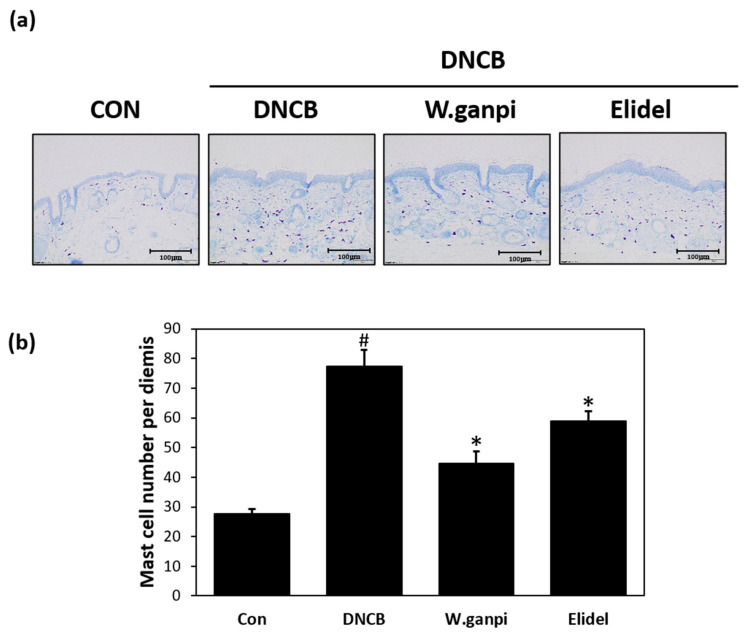
Effects of WGE on the change of the number of mast cells per dermis in AD-like symptoms induced by DNCB in hairless mice. (**a**) Results of changes in the number of the mast cells per dermis of skin lesion determined by histopathological examination. Stained by toluidine blue; (**b**) The number of mast cells per dermis. CON: control group, DNCB: DNCB-treated group, *W. ganpi*: DNCB and 1% *W. ganpi* EtOH extract-treated group, Elidel: DNCB and Elidel treated group. ^#^
*p* < 0.05 vs. the CON group; * *p* < 0.05 vs. the DNCB group.

**Figure 4 molecules-26-02016-f004:**
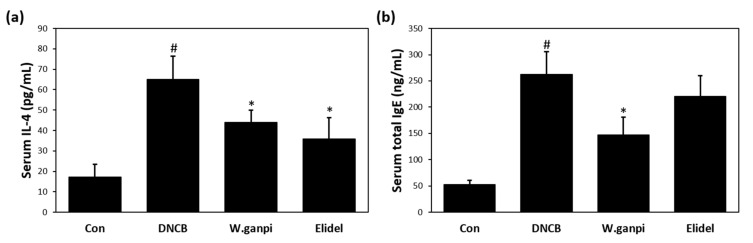
Effects of WGE on the serological change of the IL-4 and IgE concentration in AD-like symptoms induced by DNCB in hairless mice. (**a**) Serum total IgE levels; (**b**) Serum total IL-4 levels. ^#^
*p* < 0.05 vs. the CON group; * *p* < 0.05 vs. the DNCB group.

**Figure 5 molecules-26-02016-f005:**
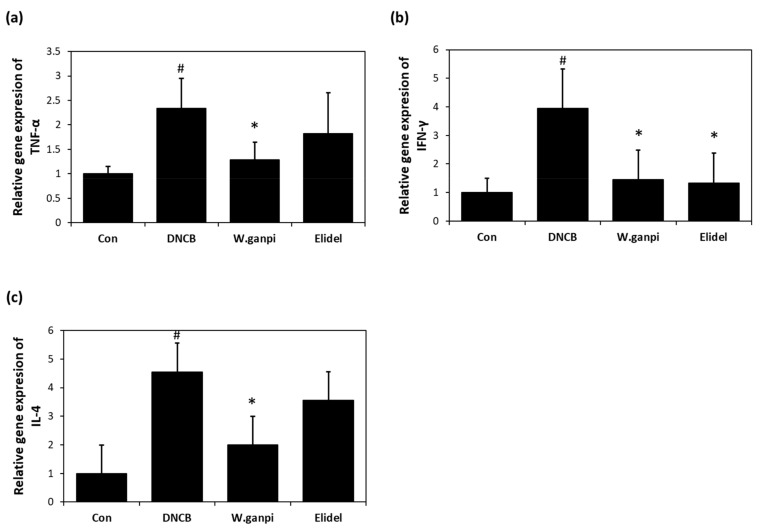
Effects of WGE on the relative gene expression of cytokines TNF-α, IFNγ, and IL-4 DNCB sensitized mouse dorsal skin. (**a**) The mRNA expression of TNF-α; (**b**) The mRNA expression of IFNγ; (**c**) The mRNA expression of IL-4 ^#^
*p* < 0.05 vs. the CON group; * *p* < 0.05 vs. the DNCB group.

**Figure 6 molecules-26-02016-f006:**
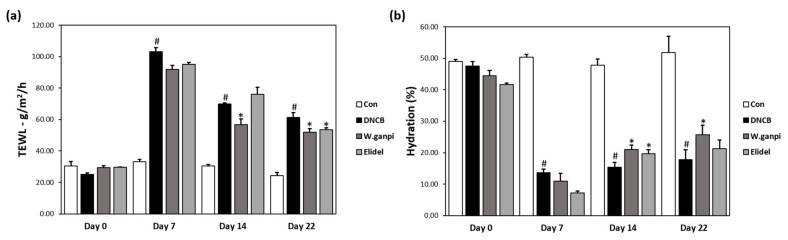
Effects of WGE on the skin barrier function in AD-like symptoms induced by DNCB in hairless mice. (**a**) Transepidermal water loss (TEWL); (**b**) Skin hydration value. ^#^
*p* < 0.05 vs. the CON group; * *p* < 0.05 vs. the DNCB group.

**Figure 7 molecules-26-02016-f007:**
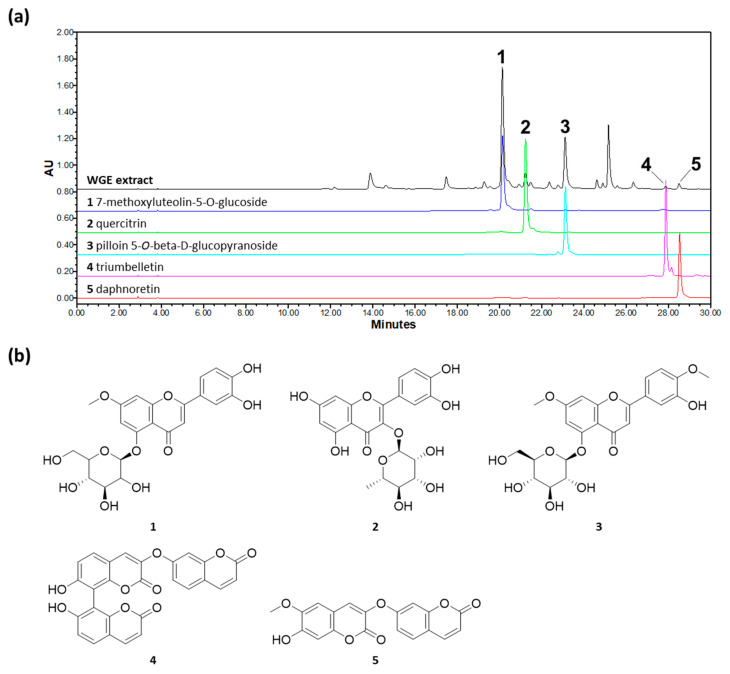
HPLC-PDA chromatograms of WGE at 340 nm (**a**) and chemical structure (**b**). Peaks: (1) 4-methoxyluteolin-5-*O*-glucoside; (2) quercitrin; (3) pilloin 5-*O*-β-d-glucopyranoside; (4) triumbelletin; (5) daphnoretin.

## Data Availability

The data presented in this study are available on request from the corresponding author.

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
