# Peer review of "Wikstroemiaganpi Extract Improved Atopic Dermatitis-Like Skin Lesions via Suppression of Interleukin-4 in 2,4-Dinitrochlorobenzene-Induced SKH-1 Hairless Mice"

_molecules, 2021, doi:10.3390/molecules26072016_

Round 1

Reviewer 1 Report

In this manuscript, the authors aimed to demonstrate the effectiveness of Wikstroemia ganpi on atopic dermatitis.  The data clearly supported this hypothesis by observing effects of ethanolic extract of W. ganpi on 2,4-dinitrochlorobenzene-induced atopic dermatitis mice.  The results are acceptable for publication.  Some suggestion is listed below for the consideration of revision.

  1. Please check if the group should be “the DNCB group” stead of “the Control group” in the following sentences.

Lines 164-165 “In the W. ganpi group, epidermal thickness was about 37.3% thinner (47.20 μm) than in the CON group (75.25 μm) (Figure 2a and 2b)” and Lines 166-168 “In the positive control-treated group (Elidel group), epidermal thickness was about 16.2% thinner (59.04 μm) than in the CON group (75.25 μm)”

  1. Please explain why you used 1%WGB? Screen from preliminary data or follow previous publication?

  1. Figure 1 showed the histology of skin. Did the authos use atopic dermatitis scores to evaluate the severity of skin?

  1. As the manuscript mentioned, skin damage was related to IL-4, released by Th2 cells.  Did the authors evaluate Th2 cells by IHC or flow cytometry?

  1. In this research, the immune response was stimulated by DNCB. Did the authors evaluate other cytokines, such as TNF-alpha, IFN-γ, and IL-4 by QPCR or Western?

  1. How did the authors identify the five compounds (three flavonoids and two coumarins) shown in Figure 6? What is the abundant peak between compounds 3 and 4?  Flavonoids are more abundant than coumarins, and which compound(s) might be responsible for the effects shown in this manuscript?  Please provide some discussion in this aspect.

Author Response

  1. Please check if the group should be “the DNCB group” stead of “the Control group” in the following sentences.

Lines 164-165 “In the W. ganpi group, epidermal thickness was about 37.3% thinner (47.20 μm) than in the CON group (75.25 μm) (Figure 2a and 2b)” and Lines 166-168 “In the positive control-treated group (Elidel group), epidermal thickness was about 16.2% thinner (59.04 μm) than in the CON group (75.25 μm)”

- I appreciate reviewer’s valuable comment. We now changed ‘in the CON group’ to ‘in the DNCB group’. Please see yellow-highlighted parts (line 184, 187, 188).

  1. Please explain why you used 1% WGB? Screen from preliminary data or follow previous publication?

- In general, it is known that the compound passes through the skin at a ratio of 1/100 to 1/1000, and thus the experiment was performed with 1% WDE (10 mg / ml), which is 1,000 times the effective concentration of 10 ug/ml in vitro.

  1. Figure 1 showed the histology of skin. Did the authors use atopic dermatitis scores to evaluate the severity of skin?

- Results of dermatitis scores are newly inserted as Figure 1c (line 112-116, 193-195). The severity of dermatitis in skin was assessed using the Eczema Area and Severity Index scoring system: 0, no symptoms; 1, mild symptoms; 2, moderate symptoms; and 3, severe symptoms. Score assessment was defined by summing dermatitis scores for erythema/hemorrhage, scarring, edema, dryness and excoriation/erosion.

  1. As the manuscript mentioned, skin damage was related to IL-4, released by Th2 cells.  Did the authors evaluate Th2 cells by IHC or flow cytometry?

- Result of IL-4 mRNA expression in the dorsal skin tissues is newly inserted as Figure 5 (line 224-231). We evaluated IL-4 using blood and tissue in this experiment, and thus didn’t evaluate Th2 cells by IHC or flow cytometry.

  1. In this research, the immune response was stimulated by DNCB. Did the authors evaluate other cytokines, such as TNF-alpha, IFN-γ, and IL-4 by QPCR or Western?

- Results of pro-inflammatory cytokines such as TNFα, IFNγ, and IL-4 are newly inserted as Figure 5 (line 133-148, 224-231). The mRNA gene expression of the pro-inflammatory cytokines was measured in mice tissues using Q-PCR. The mRNA gene expressions of TNFα, IFNγ and IL-4 were increased in the DNCB group compared to the Con group, whereas W. ganpi group is significantly decreased mRNA gene expressions in mice tissue.

  1. How did the authors identify the five compounds (three flavonoids and two coumarins) shown in Figure 6? What is the abundant peak between compounds 3 and 4?  Flavonoids are more abundant than coumarins, and which compound(s) might be responsible for the effects shown in this manuscript?  Please provide some discussion in this aspect.

- We performed a phytochemical characterization of WDE using pure compounds that we isolated from same extract. Fingerprinting result including the Rt of each compounds in newly inserted as Figure 7 (line 258-260) and UV comparisons between compounds and control are as below.

Fig. UV spectrum of WGE extract (a-e) and compounds (a’-e’). (a) 7-methoxyluteolin-5-O-glucoside; (b) quercitrin; (c) pilloin 5-O-beta-D-glucopyranoside; (d) triumbelletin; (e) daphnoretin.

The peak between compound 3 and 4 is not assigned yet. And also, in this experiment, we focused on the biological activities of extract, but not of compounds. Therefore, we couldn’t reveal which compound is responsible for the anti-AD effect of WDE. Further studies are needed to identify the active components.

Reviewer 2 Report

The work submitted by Jegal and col. Entitled “Wikstroemia ganpi extract improved atopic dermatitis-like skin lesions via suppression of interleukin-4 in 2,4-dinitrochloro-benzene-induced SKH-1 hairless mice” show the potential use of an extract of wikstroemia ganpi to reduce the skin lesions (atopic dermatitis) produced by DNCB. The experimental procedure for the determination of the grade of skin lesions and its improvement mediated by WGE shows that secondary metabolites of this plant acts decreasing the lesion induced by DNCB. Additionally, authors show that WGE is capable to maintain the epidermal functions and moisture. Inflammatory cytokines such as, IL-4 and IgE were measure and its results display a good correlation with in vivo experiments. This manuscript is an interesting work that show the potential use of natural plants in atopic dermatitis, using in vivo skin experiments, histological analysis and the quantification of inflammatory mediators.

The main problem in my chemical point of view is the analysis and determination of the flavonoids describes. How the author confirms the chemical structure of the metabolites detected by HPLC-PDA, they used pure samples of the compounds as control and they do a comparison between Tr and Abs of the mixture compounds and control? Else, some mass spectra of the different peaks are need to confirm the chemical structure of the flavonoids describes. In my opinion, it is not necessary to describe the flavonoids presents in the plant and focalize the discussion in the application of the WGE to improve AD skin lesion.

The main results of this work are the in vivo application of WGE for to treat AD, but if the author want to describe the secondary metabolites present in this extract, they must improve the chemical analysis including at least mass spectrometry.

In line 87, author must to describe if they use a dry sample of plant or not (21 g of dry plant)

I recommend this work to publication but author must improve and/or define if they want to describe the secondary metabolites present in the WGE.

Author Response

The main problem in my chemical point of view is the analysis and determination of the flavonoids describes. How the author confirms the chemical structure of the metabolites detected by HPLC-PDA, they used pure samples of the compounds as control and they do a comparison between Tr and Abs of the mixture compounds and control? Else, some mass spectra of the different peaks are need to confirm the chemical structure of the flavonoids describes. In my opinion, it is not necessary to describe the flavonoids presents in the plant and focalize the discussion in the application of the WGE to improve AD skin lesion.

The main results of this work are the in vivo application of WGE for to treat AD, but if the author want to describe the secondary metabolites present in this extract, they must improve the chemical analysis including at least mass spectrometry.

- I appreciate reviewer’s valuable comment. We performed a phytochemical characterization of WDE using pure compounds that we isolated from same extract. Fingerprinting result including the Rt of each compounds in newly inserted as Figure7 (line 258-260) and UV comparisons between compounds and control are as below.

Fig. UV spectrum of WGE extract (a-e) and compounds (a’-e’). (a) 7-methoxyluteolin-5-O-glucoside; (b) quercitrin; (c) pilloin 5-O-beta-D-glucopyranoside; (d) triumbelletin; (e) daphnoretin.

In line 87, author must to describe if they use a dry sample of plant or not (21 g of dry plant)

- It is now changed to ‘21 g of dry plant’ (line 87).

I recommend this work to publication but author must improve and/or define if they want to describe the secondary metabolites present in the WGE.

- We improved this revised version of manuscript by newly inserting several biological data such as dermatitis score (line 112-116, 193-195) and pro-inflammatory cytokines (TNFα, IFNγ, and IL-4) (line 133-148, 224-231). In this experiment, we focused on the biological activities of extract, but not of compounds. Therefore, we couldn’t reveal which compound is responsible for the anti-AD effect of WDE. Further studies are needed to identify the active components.

The main problem in my chemical point of view is the analysis and determination of the flavonoids describes. How the author confirms the chemical structure of the metabolites detected by HPLC-PDA, they used pure samples of the compounds as control and they do a comparison between Tr and Abs of the mixture compounds and control? Else, some mass spectra of the different peaks are need to confirm the chemical structure of the flavonoids describes. In my opinion, it is not necessary to describe the flavonoids presents in the plant and focalize the discussion in the application of the WGE to improve AD skin lesion.

The main results of this work are the in vivo application of WGE for to treat AD, but if the author want to describe the secondary metabolites present in this extract, they must improve the chemical analysis including at least mass spectrometry.

- I appreciate reviewer’s valuable comment. We performed a phytochemical characterization of WDE using pure compounds that we isolated from same extract. Fingerprinting result including the Rt of each compounds in newly inserted as Figure7 (line 258-260) and UV comparisons between compounds and control are as below.

Fig. UV spectrum of WGE extract (a-e) and compounds (a’-e’). (a) 7-methoxyluteolin-5-O-glucoside; (b) quercitrin; (c) pilloin 5-O-beta-D-glucopyranoside; (d) triumbelletin; (e) daphnoretin.

In line 87, author must to describe if they use a dry sample of plant or not (21 g of dry plant)

- It is now changed to ‘21 g of dry plant’ (line 87).

I recommend this work to publication but author must improve and/or define if they want to describe the secondary metabolites present in the WGE.

- We improved this revised version of manuscript by newly inserting several biological data such as dermatitis score (line 112-116, 193-195) and pro-inflammatory cytokines (TNFα, IFNγ, and IL-4) (line 133-148, 224-231). In this experiment, we focused on the biological activities of extract, but not of compounds. Therefore, we couldn’t reveal which compound is responsible for the anti-AD effect of WDE. Further studies are needed to identify the active components.

Reviewer 3 Report

This article describes the usefulness of ethanolic extract of W. ganpi plants in relieving chemically-induced atopic dermatitis (AD) in mice. The authors showcased the therapeutic effect through the severity of AD appearance, epidermal thickness, mast cells infiltration, IL4 & IgE serum level, and transepidermal water loss. Moreover, the authors characterized the compounds contained in the extract through chromatography. 

Overall, I believe this article is of significant quality for publication in Molecules, and would recommend its acceptance following several clarification/revisions below:

  • In Fig 1B, the severity of AD at d7 for DNCB group seems greater than that of Elidel group. Can the authors clarify whether any scoring/assessment was done to ensure comparable AD severity before treatment?
  • If not, I would suggest to adopt & show a scoring system to ensure comparable induction of AD at day 7. Also, it can help readers to understand the alleviated AD symptoms mentioned in page 4 line 160
  • Related to above, the difference between W.ganpi & DNCB group at day 14 is already unclear. I strongly recommend the authors to show image during earlier treatment, maybe at day 9 or 12.
  • In page 4 line 165-168, data shows 75.25um for DNCB group instead of CON group. I believe the authors made an error in their comparisons "W.ganpi & Elidel group thinner than CON group". Please clarify.
  • Similarly, please check and clarify the comparisons mentioned for Fig 3 result (Mast cell infiltration).

Author Response

In Fig 1B, the severity of AD at d7 for DNCB group seems greater than that of Elidel group. Can the authors clarify whether any scoring/assessment was done to ensure comparable AD severity before treatment? If not, I would suggest to adopt & show a scoring system to ensure comparable of AD at day 7. Also, it can help readers to understand the alleviated AD symptoms mentioned in page 4 line 160.

Related to above, the difference between W.ganpi & DNCB group at day 14 is already unclear. I strongly recommend the authors to show image during earlier treatment, maybe at day 9 or 12.

- I appreciate reviewer’s valuable comment. Results of dermatitis scores are newly inserted as Figure 1c (line 112-116, 193-195) as below. The severity of dermatitis in skin was assessed using the Eczema Area and Severity Index scoring system: 0, no symptoms; 1, mild symptoms; 2, moderate symptoms; and 3, severe symptoms. Score assessment was defined by summing dermatitis scores for erythema/hemorrhage, scarring, edema, dryness and excoriation/erosion. It takes at least 7 days to show in vivo effect, thus the sample might not be effective at day 14.

In page 4 line 165-168, data shows 75.25um for DNCB group instead of CON group. I believe the authors made an error in their comparisons "W.ganpi & Elidel group thinner than CON group". Please clarify. Similarly, please check and clarify the comparisons mentioned for Fig 3 result (Mast cell infiltration).

- We now changed ‘in the CON group’ to ‘in the DNCB group’. Please see yellow-highlighted parts (line 184, 187, 188).

In Fig 1B, the severity of AD at d7 for DNCB group seems greater than that of Elidel group. Can the authors clarify whether any scoring/assessment was done to ensure comparable AD severity before treatment? If not, I would suggest to adopt & show a scoring system to ensure comparable of AD at day 7. Also, it can help readers to understand the alleviated AD symptoms mentioned in page 4 line 160.

Related to above, the difference between W.ganpi & DNCB group at day 14 is already unclear. I strongly recommend the authors to show image during earlier treatment, maybe at day 9 or 12.

- I appreciate reviewer’s valuable comment. Results of dermatitis scores are newly inserted as Figure 1c (line 112-116, 193-195) as below. The severity of dermatitis in skin was assessed using the Eczema Area and Severity Index scoring system: 0, no symptoms; 1, mild symptoms; 2, moderate symptoms; and 3, severe symptoms. Score assessment was defined by summing dermatitis scores for erythema/hemorrhage, scarring, edema, dryness and excoriation/erosion. It takes at least 7 days to show in vivo effect, thus the sample might not be effective at day 14.

In page 4 line 165-168, data shows 75.25um for DNCB group instead of CON group. I believe the authors made an error in their comparisons "W.ganpi & Elidel group thinner than CON group". Please clarify. Similarly, please check and clarify the comparisons mentioned for Fig 3 result (Mast cell infiltration).

- We now changed ‘in the CON group’ to ‘in the DNCB group’. Please see yellow-highlighted parts (line 184, 187, 188).

Round 2

Reviewer 2 Report

I agree with the changes made by the authors